# Assessment of Air Quality and Meteorological Changes Induced by Future Vegetation in Madrid

**David de la Paz** [1], **Juan Manuel de Andrés** [1], **Adolfo Narros** [1], **Camillo Silibello** [2], **Sandro Finardi** [2], **Silvano Fares** [3,4], **Luis Tejero** [5], **Rafael Borge** [1,*] and **Mihaela Mircea** [6]

1   Department of Chemical and Environmental Engineering, ETSII—Universidad Politécnica de Madrid (UPM), 28040 Madrid, Spain; david.delapaz@upm.es (D.d.l.P.); juanmanuel.deandres@upm.es (J.M.d.A.); adolfo.narros@upm.es (A.N.)
2   ARIANET, 20159 Milán, Italy; c.silibello@aria-net.it (C.S.); s.finardi@aria-net.it (S.F.)
3   Research Centre for Forestry and Wood, Council for Agricultural Research and Economics (CREA), 00184 Rome, Italy; silvano.fares@cnr.it
4   Institute of BioEconomy, National Research Council of Italy (CNR), 00185 Rome, Italy
5   Ayuntamiento de Madrid, 28014 Madrid, Spain; tejeroel@madrid.es
6   Laboratory of Atmospheric Pollution, Italian National Agency for New Technologies, Energy and Sustainable Economic Development—ENEA, 40129 Bologna, Italy; mihaela.mircea@enea.it
*   Correspondence: rafael.borge@upm.es

**Abstract:** Nature-based solutions and green urban infrastructures are becoming common measures in local air quality and climate strategies. However, there is a lack of analytical frameworks to anticipate the effect of such interventions on urban meteorology and air quality at a city scale. We present a modelling methodology that relies on the weather research and forecasting model (WRF) with the building effect parameterization (BEP) and the community multiscale air quality (CMAQ) model and apply it to assess envisaged plans involving vegetation in the Madrid (Spain) region. The study, developed within the VEGGAP Life project, includes the development of two detailed vegetation scenarios making use of Madrid's municipality tree inventory (current situation) and future vegetation-related interventions. An annual simulation was performed for both scenarios (considering constant anthropogenic emissions) to identify (i) variations in surface temperature and the reasons for such changes, and (ii) implications on air-quality standards according to EU legislation for the main pollutants ($PM_{10}$, $PM_{2.5}$, $NO_2$ and $O_3$). Our results suggest that vegetation may have significant effects on urban meteorology due to changes induced in relevant surface properties such as albedo, roughness length or emissivity. We found a net-heating effect of around +0.18 °C when trees are introduced in dry, scarcely vegetated surfaces in the city outskirts. In turn, this enhances the planetary boundary layer height (PBLH), which brings about reductions in ambient concentrations of relevant pollutants such as $NO_2$ (in the range of 0.5–0.8 µg m$^{-3}$ for the annual mean, and 2–4 µg m$^{-3}$ for the 19th highest 1 h value). Conversely, planting new trees in consolidated urban areas causes a cooling effect (up to −0.15 °C as an annual mean) that may slightly increase concentration levels due to less-effective vertical mixing and wind-speed reduction caused by increased roughness. This highlights the need to combine nature-based solutions with emission-reduction measures in Madrid.

**Keywords:** air quality; urban vegetation; nature-based solutions; meteorology; VEGGAP project





## 1. Introduction

Climate change impacts are expected to increase in severity and frequency in the coming decades [1–3]. In this context, nature-based solutions (NBS) may contribute to reducing risk related to climate change and restoring and protecting ecosystems, while providing multiple additional benefits (environmental, social and economic) [4,5].

According to [6], the concept of NBS lies on the boundaries between science and policy. Even though these solutions have not yet been widely adopted in policy, planning, and

governance so far [7], in recent years they have been becoming increasingly popular in urban strategies to deal with the environmental dysfunctions of densely populated areas that have negative health effects on urban populations [8,9].

The role of nature supporting the economy as well as the livelihood of citizens has been widely described in the scientific literature. According to [10], several main NBS actions are recommended to be taken forward: urban regeneration [11], coastal resilience [12,13], multifunctional nature-based watershed management and ecosystem restoration [14], increasing the sustainable use of matter and energy as well as carbon sequestration [15], fostering outdoor activities for mental and physical health [16,17], enhancing the insurance value of ecosystems and improving wellbeing in urban areas [18,19].

As to this last point, the introduction of vegetation in cities has associated benefits regarding climate-change mitigation, such as an increase in street shadows or a decrease in the heat-island effect, that are of essential importance for the health and living comfort of city dwellers [20,21]. Vegetation in cities has also been described as a way to mitigate pollution thanks to enhanced wet- and dry-deposition processes [22–24]. However, vegetation is also a source of emissions for some substances, such as pollen and ozone, and particulate matter precursors that can affect negatively people's health [25–27]. These negative impacts of urban ecosystems on urban dwellers are defined as ecosystem disservices (EDS) [28].

Vegetation is the main source of atmospheric volatile organic compounds (VOCs), accounting for about 90% of the total emissions worldwide [29]. Biogenic VOC (BVOCs), dominated by isoprene and monoterpenes, are an important atmospheric constituent affecting both the gas phase and heterogeneous chemistry of the troposphere [30], as well as being important precursors for the formation of ozone and secondary organic aerosols (SOA) [31–33], which may exacerbate respiratory conditions such as rhinitis and asthma [34,35].

However, the effect of vegetation on local air quality is very complex and depends on many more factors than just emissions. Urban vegetation modifies the local meteorology by affecting air temperature and relative humidity [36–38], altering the wind direction and wind speed due to changes in terrain roughness [39–41], and changing the thermal properties of the surface, such as surface albedo, surface humidity, etc. [42]. All these factors modify the boundary layer height (BLH) that controls the vertical mixing of urban pollutants and, therefore, their corresponding concentration levels [43].

Therefore, it is necessary to develop assessment tools that allow a consistent analysis of the air-quality effects associated with the implementation of green infrastructures in the design of plans and measures involving vegetation in urban areas.

To date, there are few studies in the scientific literature that delve into the impact of NBS on air quality, and most of them address the aerodynamic and deposition effects of vegetation on street/neighborhood ventilation and on pollutant concentration by means of computational fluid dynamic (CFD) simulations [44–48]. While these studies provide valuable information on the impact of vegetation at a microscale, they are not able to describe the complex chemical and physical atmospheric processes that account for the formation of secondary pollutants. The effects of city-wide NBS transcend the microscale level, so they must be studied from a broader, mesoscale approach [49]. The work presented here, developed within the VEG-GAP Life project (https://www.lifeveggap.eu, accessed 20 april 2022), aims at consistently quantifying the overall effects of future vegetation-related interventions in or around the city of Madrid on local meteorology and air quality, including secondary pollutants, through mesoscale simulations.

## 2. Materials and Methods

### 2.1. Description of the Model System

This study relies on a state-of-the-science modelling system that includes three main components: (i) a meteorological model, (ii) an emission model and (iii) a chemical-transport model. To analyze the impact of vegetation on meteorology and to provide the relevant inputs for the chemical-transport model, we applied WRFv4.1.2 (weather

research and forecasting) [50], coupled with the building energy parametrization (BEP) [51]. Physical options and parametrizations are shown in Table 1. They were selected according to previous sensitivity analyses specifically made for this modelling domain [52,53]. Of note, we selected the PBL scheme that has a better compatibility with BEP and it is thus recommended by the developers of this urban parameterization [51].

**Table 1.** WRF model setup.

| Option | Setup |
| --- | --- |
| Initialization | ERA5 |
| Shortwave radiation | MM5 |
| Longwave radiation | GFDL |
| Land-surface model | Noah LSM |
| Microphysics scheme | WSM6 |
| PBL scheme | Boulac |
| Surface layer option | Monin–Obukhov |
| Cumulus parametrization | No |
| Urban physics | BEP (building energy parameterization) |
| Nudging | No |

We performed a comprehensive statistical assessment of WRF outputs for the complete year 2015 (baseline scenario for the VEGGAP project). As summarized in Tables S1 and S2 in the Supplementary Material, we found a satisfactory model performance for a mesoscale model, suggesting that our results are reliable enough to support the analysis proposed.

Anthropogenic emissions were processed through SMOKEv3.6.5 (Sparse Matrix Operator Kernel System) [54], specifically adapted for the Iberian Peninsula [55,56]. The estimation of emissions from the vegetation relies on PSEM (plant-specific emission model) [57] that includes specific basal emission factors for a wide range of tree species that are modified according to concurrent temperature, solar radiation and seasonality for each meteorological scenario. This allows a consistent representation of biogenic VOCs and meteorological information used to feed the chemical-transport model CMAQv5.3.2 [58,59]. Through a nested domain approach [60] CMAQ provides a consistent representation of physical transport and deposition phenomena, as well as chemical reactions in the atmosphere. In this study, they were represented by the carbon bond (CB6) chemical mechanism [61] and the aerosol module AERO6 [62]. This modelling approach allows a consistent estimation of all physical and chemical properties of atmosphere. Following the objectives of the VEG-GAP project, we focus the analysis on $NO_2$, PM and $O_3$, which are the most relevant pollutants in Madrid [63].

### 2.2. Modelling Domains

The nested modelling domains used in this study are shown in Figure 1. The mother domain (D1) covers the whole of Europe and Northern Africa with a spatial resolution of $27 \times 27$ km and provides dynamic boundary conditions for D2, which includes the Iberian Peninsula, with a spatial resolution of $9 \times 9$ km. D3, with a spatial resolution of $3 \times 3$ km, focuses on the central area of Spain; additionally, finally, the innermost modelling domain focuses on the Madrid Region, with a spatial resolution of $1 \times 1$ km. The results and discussion are based on the model outputs for this domain, which consists of $136 \times 144$ grid cells in the east—west direction and the north—south direction, respectively. As shown in Table S2 (Supplementary Material), the modelling domains used for WRF are slightly larger, to avoid boundary artifacts. The vertical structure is identical for all 4 modelling domains and it includes 38 layers entirely covering the troposphere. Of note, we include 18 layers within the first kilometer (the lower vertical level has a 7 m height) above the ground to allow a detailed representation of relevant atmospheric processes within the planetary boundary layer.

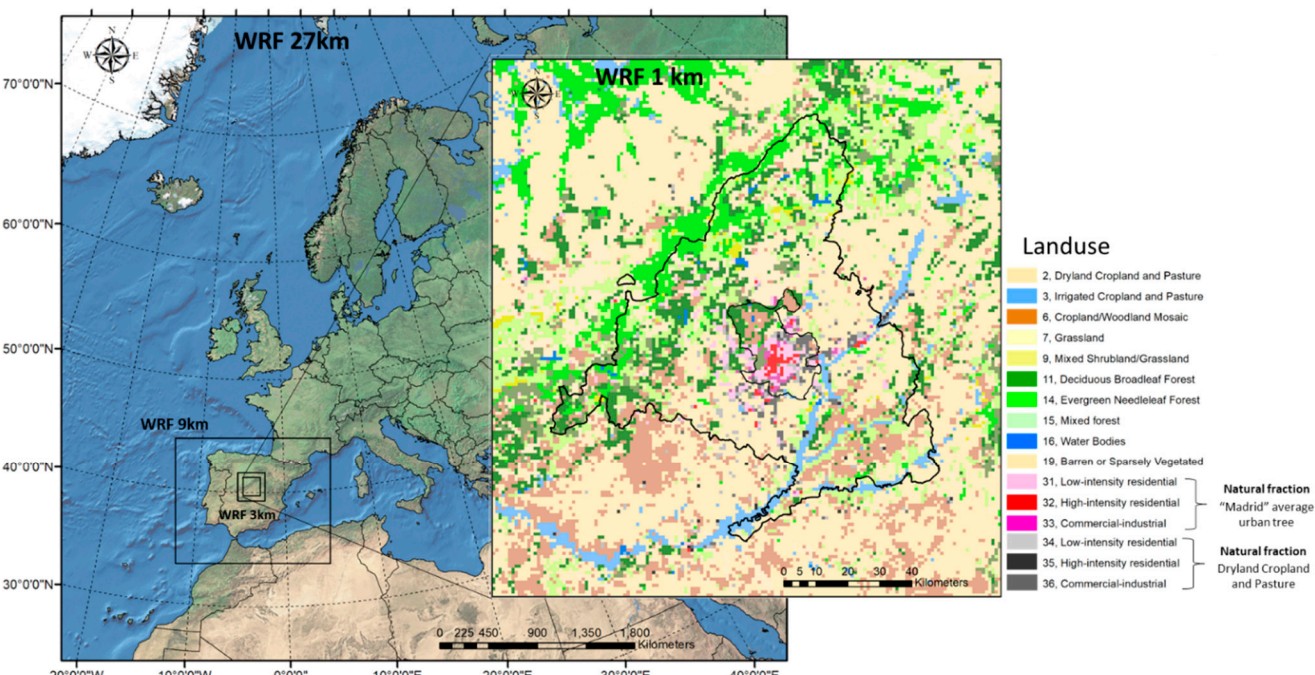

**Figure 1.** Nested modelling domains and detail of the innermost 1 km² resolution domain showing dominant land-use types.

The temporal domain of our simulations covers the entire year 2015, so we can capture the effect of vegetation under a wide range of meteorological conditions and phenologies. In addition, this allows us to assess the impact of vegetation on air-quality standards according to the Air Quality Directive 2008/50/EC, such as annual limit values for $NO_2$, $PM_{10}$, $PM_{2.5}$ and $O_3$.

### 2.3. Vegetation Scenarios

We assessed the impact of vegetation-related interventions on meteorology and air quality through the comparison of two scenarios (both annual runs):

- Baseline: intended to represent the current situation (year 2015);
- Future: incorporates the vegetation associated to intended measures in the whole Madrid Region.

Surface properties in a mesoscale meteorological model, such as WRF, are represented by average biophysical properties of different land uses that govern the exchange of heat and momentum between the atmosphere and the ground. Current land uses were taken from the Corine land cover (CLC 2012) and mapped to USGS classes. The details of this procedure can be found elsewhere [53]. These include natural and urban classes, where built up areas are significant. In the second case, grid cells may include some vegetation too and their influence on such properties needs to be considered as well. The part of the modelling system that solves the interactions between the atmosphere and the ground depending on the land-use type (Figure 1) is the land-surface model (LSM), a particularly important feature for this study. We relied on the Noah land-surface model (LSM) [64], which has been identified as the best-performing option for our modelling domain in previous studies [52,53]. However, the Monin–Obukhov similarity theory (MOST) behind LSM in mesoscale models does not hold on urban areas, so the multilayer urban parameterization BEP was applied for our simulations in D4. This urban canopy parameterization considers 3 urban land use classes: 'Low Intensity Residential', LIR (31), 'High Intensity Residential', HIR (32) and 'Commercial & Industrial', C&I (33). The user of the standard model defines an average urban fraction, UF (ranging from 0 to 1), for each of these categories, representing the fraction of the surface covered by artificial

materials (buildings, roads and sidewalks). In the present study, we assigned UF for each grid cell using the variable FRC_URB2D. The remaining area (1-UF) within urban grid cells is addressed as natural fraction and assigned the same physical properties (those of 'Cropland/grassland mosaic', default land-use type). Using a single land use to represent the non-urban fractions in built areas is a strong simplification, since important properties such as leaf area index (LAI), albedo, canopy height or roughness length may vary largely depending on the actual vegetation in different grid cells within the city.

In order to increase the flexibility of this modelling framework and to describe more realistically different interventions related to urban vegetation, we introduced some modifications in the code. Specifically, we added 6 new urban land uses (34 to 39) to allow the model to map the natural-fraction properties (that apply to 1-UF fraction of each grid cell) to 6 different natural-land uses that may be present in the city.

### 2.3.1. Baseline Scenario

This scenario includes 6 urban land-use types (31 and 34 for LIR, 32 and 35 for HIR and 33 and 36 for C&I). The difference between the different categories for a given urban morphology is the vegetation type present in the non-urban fraction. To represent this, we made use of Madrid's municipality tree inventory, which includes detailed information for each tree in the city (location, species, height, etc.). We analyzed this high-resolution urban vegetation cover to define the 'average urban tree' in Madrid, which was assigned relevant properties according to the values provided by the Noah LSM. The 'average urban tree' in Madrid is 85% 'Deciduous Broadleaf Forest', 13% 'Evergreen Needleleaf Forest' and 2% 'Evergreen Broadleaf Forest'. Weighted properties (Table S4) were assigned to the natural fraction of land uses 31, 32 and 33 according to these percentages. For the natural fraction of other urban categories (34, 35 and 36), the properties of the 'Dryland, Cropland and Pasture' land-use class were assigned (Table S4).

### 2.3.2. Future Scenario

We built a Future Scenario that intends to represent all the relevant interventions envisaged in urban plans and strategies for Madrid in the next decades (Figure 2). The 4 major naturalization measures included in this scenario are as follows:

1. 'Arco verde': linear plantations along drovers' roads, trails and other rural paths to connect already existing periruban forests or green areas, a number or native species are considered.
2. 'Barrios productores': areas designated for the municipal network of urban vegetable gardens, considered as low-density shrub areas.
3. 'Madrid Nuevo Norte': a major urban development approved by the local and regional governments. It is intended to be a carbon-neutral mix of uses including new residential, business and green areas. Lacking more specific information, we have assumed that this green area will have the average characteristics (in term of species and density) of existing parks in Madrid.
4. 'Bosque metropolitano': The metropolitan forest is the most ambitious action of NBS projected for the coming years in the city of Madrid. It is framed on the air-quality and climate-change plans and the sustainability and urban green infrastructure strategies. This action aims to create a 75 km periurban green ring, which will connect existing parks, gardens and natural spaces and expand the urban green areas by more than 5000 hectares.

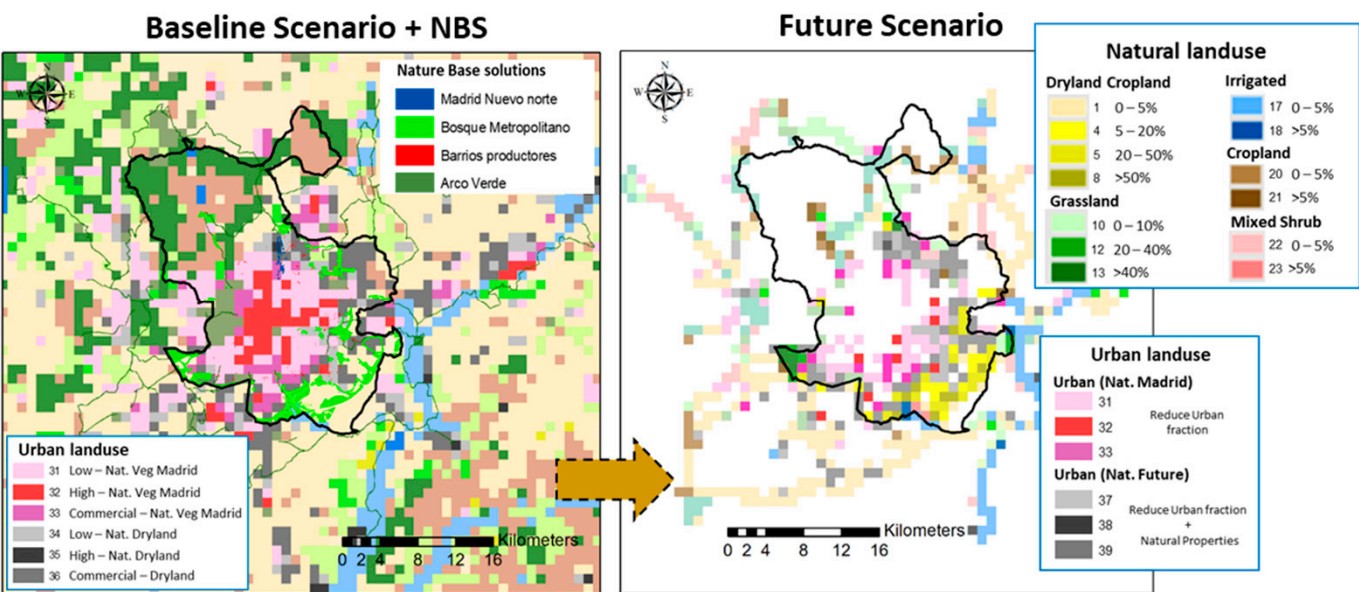

**Figure 2.** Baseline scenario along with the NBS-based interventions envisaged in the Future Scenario (**left**). Model grid cells affected by those vegetation changes and land-use category assigned in the Future Scenario (**right**).

Since the project was in definition stage, the allocation and exact selection of tree species was not completely defined at the moment of performing our simulations; therefore, using the existing information as a base, we had to make some assumptions regarding trees and shrubs species and plantation density, according to the soil types, topography, etc., of each intervention area.

All combined, 724 grid cells were affected by vegetation changes in the Future Scenario. A total of 65% of them are non-urban land uses, mostly 'Dryland Cropland and Pasture' (42%) and 'Grassland' (13%), while the remaining 35% are urban areas. For the first group, each grid cell was re-classified into 4 new natural-land uses. Grid cells formerly not wooded were assigned to classes not present in our modelling domain. While all of them assume the average properties for new trees, they differ in the surface within the grid cell affected by reforestation (0%–5%, 5%–20%, 20%–50% and >50%, respectively). Physical properties (albedo, canopy height, roughness length, etc.) were computed as weighted averages using these percentages (Table S4). Figure 3 provides the example of a grid cell affected by the 'Bosque metropolitano'.

As for urban grid cells where new vegetation is introduced (35%), we considered 2 different cases. For grid cells labelled as 31 (LIR), 32 (HIR) or 33 (C&I), urban classes that already contain trees, we simply reduced the urban fraction (UF), keeping vegetation properties unchanged. This is the case of 5% of urban grid cells affected by future interventions. For those urban land uses that do not contain trees (classes 34 (LIR), 35 (HIR) and 36 (C&I)), in addition to the corresponding change of the UF, the properties of the natural fraction (1-UF) were changed to match those of the vegetation being introduced. New land-use categories, 37, 38 and 39, were assigned to urban grid cells whose natural fraction was 'Dryland Cropland and Pasture' before the intervention, depending on the percentage of area affected by tree plantation. For simplicity, we assumed the average properties to be representative of 'Bosque metropolitan,' since it is the most important intervention (88% of the total area of future revegetation actions in Madrid). This corresponds to 90% 'Dryland Cropland and Pasture', 7% 'Evergreen Needleleaf Forest' and 3% 'Deciduous Broadleaf Forest'.

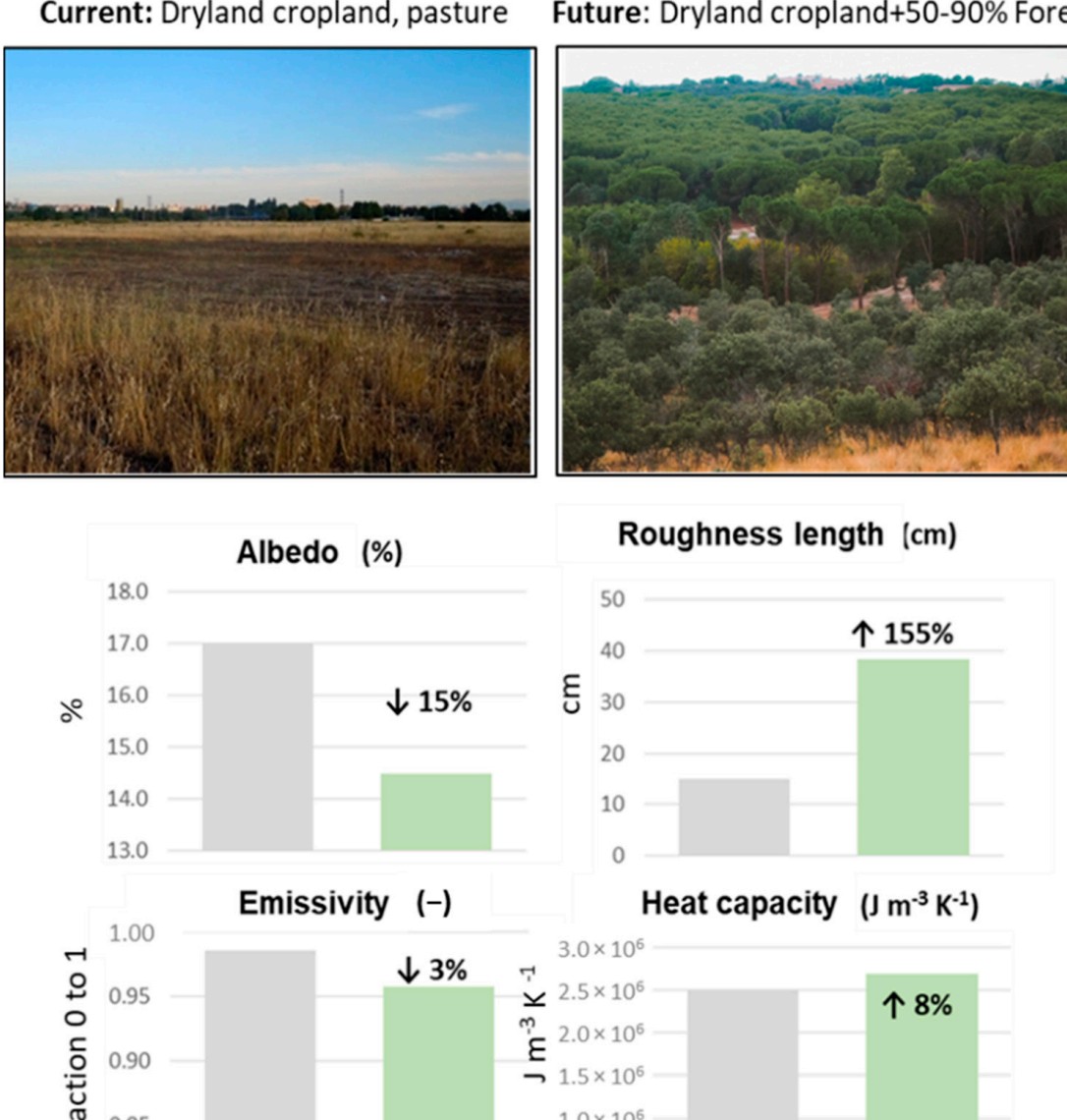

**Figure 3.** Values of albedo, roughness length, emissivity and heat capacity for current (grey bars) and future (green bars) scenarios for a grid cell where new trees ('Bosque Metropolitano') are introduced.

While BVOC emissions were provided by PSEM (considering species-specific emission rates), using scenario-specific meteorology, we kept anthropogenic emission constant for both scenarios. Although this may be rather unrealistic for the Future Scenario, we assumed this experimental design to be able to isolate the specific impact of vegetation for a given level of emissions.

## 3. Results and Discussion

First, we present the impact of vegetation on urban meteorology, focusing on temperature, PBLH and wind speed as relevant variables for air quality [65,66]. Then, we summarize our findings regarding $NO_2$, $O_3$, $PM_{10}$, and $PM_{2.5}$ ambient concentration levels. We discuss the results averaged over the complete year as well as the winter (21 December to 21 March) and summer (21 June to 21 September) periods, to gain a better perspective on seasonal variability.

*3.1. Impact of NBS on Meteorology*

Figure 4a presents the annual mean temperature at ground level (T2) for the Baseline Scenario. This image clearly shows the heat-island effect of Madrid city in the center of the modeling domain. Topography also plays an important role, explaining the large temperature differences between the northern limit of the Madrid Region (6–10 °C) that corresponds to the 'Sierra de Guadarrama' mountain range (2000–2400 m height above sea level) and the southern part of the modeling domain, with annual mean temperatures up to 17 °C in the 'Tajuña' and 'Henares' valleys. Figure 4b–d show the difference between the Future Scenario and the Baseline Scenario for T2 as an annual mean as well as the summer and winter mean, respectively. Positive values (>0), represented in red, correspond to increments, while negative values (in blue) indicate a decrease induced by future NBS (represented in light green). In all three figures, we observe positive values in the close vicinity of interventions. Of note, the introduction of trees in barren areas brings about air temperature increases up to 0.20 °C (Table 2) (1%, Figure S1). This increase may be related to augmented sensible-heat (SH) flux in the areas where new trees are included (Figure S2), due to a decrease in albedo and enhanced energy absorption (Figure S3). Overall SH fluxes are considerably higher than latent heat (LH), something typical of relatively dry soils [67,68].

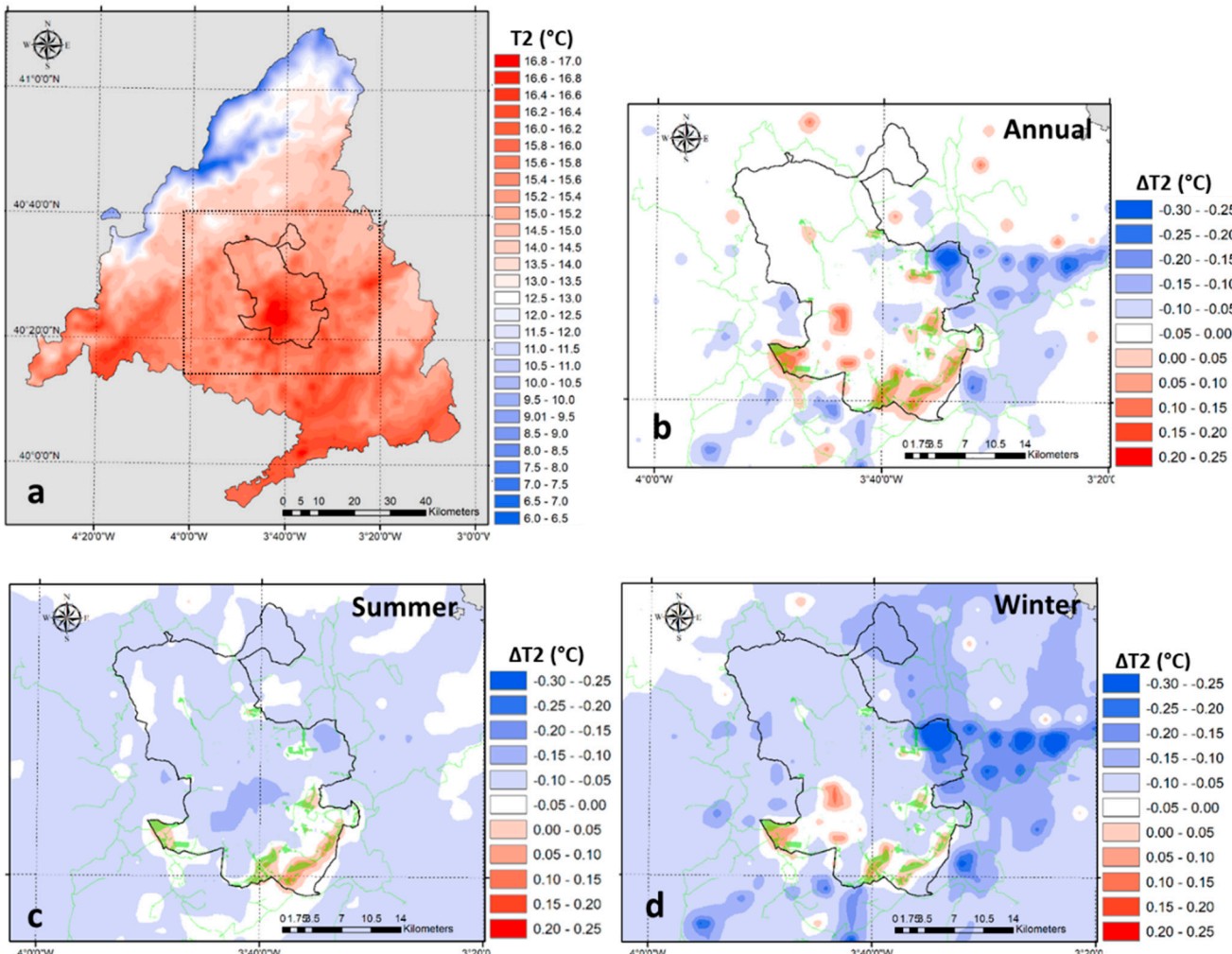

**Figure 4.** Annual mean temperature for the Baseline Scenario (**a**). Expected variation in temperature: annual mean (**b**), summer mean (**c**) and winter mean (**d**), due to future vegetation.

**Table 2.** Summary of the effect of the NBS on the main meteorological variables (relative change in brackets).

| Statistic | T (2 m) (°C) | WS (10 m) (m s$^{-1}$) | PBL Height (m) |
|---|---|---|---|
| **Maximun** | 0.20, (0.9%) | 0.25, (17.3%) | 43.4, (7.1%) |
| **Minimun** | −0.32, (−2.0%) | −0.40, (−17.2%) | −65.5, (−10.9%) |
| **Average** | −0.03, (−0.2%) | −0.03, (−1.0%) | −5.6, (−1.0%) |

Although T2 increases over more extended areas during winter (Figure 4), increased evapotranspiration in summer strengthens the heat transfer from the atmosphere to the surface and may offset upward fluxes, causing a net cooling effect [69].

On the other hand, increasing the vegetation within the city (e.g., the area affected by 'Madrid Nuevo Norte') produces reductions in annual average temperature of up to 0.15 °C, due to lower energy absorption rates (higher albedo and lower heat capacity) along with increased evapotranspiration.

In addition to local changes, we observe temperature variations in areas not directly affected by the introduction of vegetation, such as the Henares and Jarama valley (northeast of our modelling domain, Figure S4). The cooling effect in that area is presumably related to changes in wind patterns, both in speed (Figure 5b) and direction (Figure S5), that lead to cooler air advection and alter surface heat balance.

Annual average changes in wind speed (at a 10 m height) are shown in Figure 5a. The model outputs from the baseline simulations clearly show that wind speed within Madrid City is 2–3 m s$^{-1}$ lower than that of nearby rural areas. In contrast, the highest average values are found in the mountain range up north, with annual average values up to 7.5 m s$^{-1}$. The impact of the envisaged NBS on those average values is presented in Figure 5b. Our model results suggest that new vegetation may considerably reduce wind speed (blue tones), of up to 0.4 m s$^{-1}$ (Table 2) in the locations where more substantial interventions are planned, such as in the southeastern area of the modelling domain. As shown in Figure S1, in relative terms, annual average wind speed shows 5%–10% reductions in most of the urban area. This seems to be related to dramatic changes in the roughness length (e.g., Figure 3) in previously unwooded natural areas (Figure 2). As a consequence, we observe friction velocity (a turbulence scale) increases of up to 0.2 m s$^{-1}$. Although mainly evergreen species are considered in the reforestation of 'Bosque Metropolitano' (*Quercus ilex*, *Pinnus Pinea* and *Pinus halepensis* predominantly), this effect is stronger in summer (Figure S5), when maximum LAI values are reached [70]. The results shown in Figure 4 suggest that the effect on a given location may depend on mesoscale phenomena, since wind speed changes can be seen in areas not directly affected by NBS. Increased friction and reduced advection may cause wind speed increases in the surroundings of interventions, an effect that may be modulated by regional circulation associated to topographical features. In this particular case, we observe increases in wind speed of up to 0.3 m s$^{-1}$ in winter in the Henares and Jarama Valleys (Figure S4).

Our Future-Scenario simulation also indicates that new vegetation may induce changes in wind directions, more evidently during summer (Figure S5), of up to 10° as an average in summer. Wind shifts concentrate on the Jarama and Henares valleys (east and northeast of our modelling domain), suggesting that local topography is an important factor for the impact of NBS on meteorology.

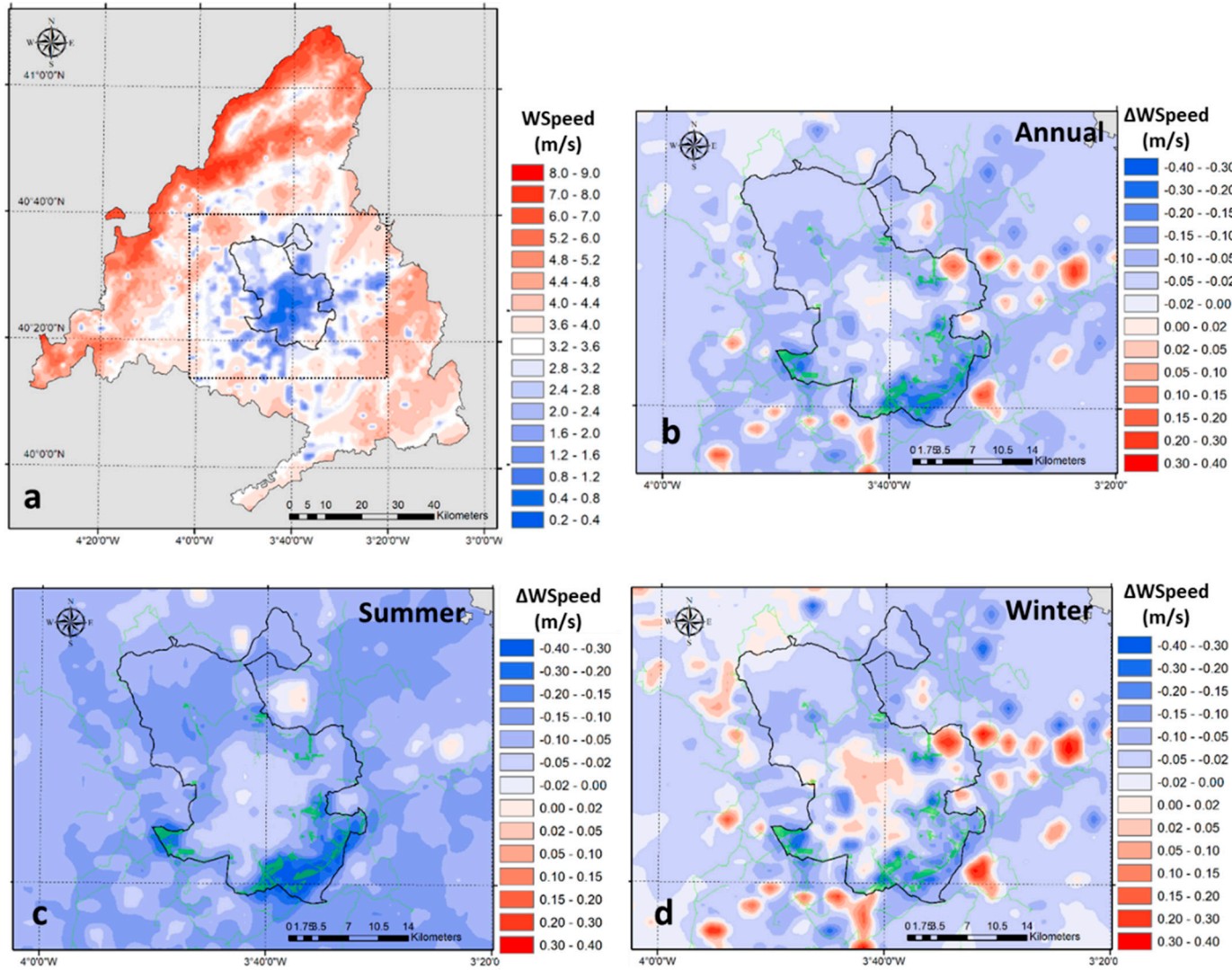

**Figure 5.** Annual mean wind speed (10 m) for the Baseline Scenario (**a**). Expected variation in wind intensity: annual mean (**b**), summer mean (**c**) and winter mean (**d**), due to future vegetation.

Finally, we show the impact of planned NBS on the PBLH, identified as a critical variable for ambient concentration levels in this particular area in previous studies [66]. This meteorological variable is closely related to the amount of turbulence, both thermal and mechanical, generated by the surface. Consequently, it is clear in Figure 6a that, currently, average PBLH is considerably higher in built-up areas, with mean values in downtown Madrid of up to 800 m (around 600 m in winter and 1000 m in summer). The changes from the baseline scenario are also dependent on changes in vertical heat fluxes and mechanical turbulence. We observe that PBLH changes are dominated by convective processes and, thus, it matches the changes described for T2. Consequently, we observe a generalized drop, of up to 65 m of PBLH in summer (Figure 6c), and increases in winter (of up to 40 m) in the close vicinity of the main interventions. While the variations are similar in absolute terms, they are relatively more important in winter (8% in comparison to a maximum value of 4% in summer, as shown in Figure S1).

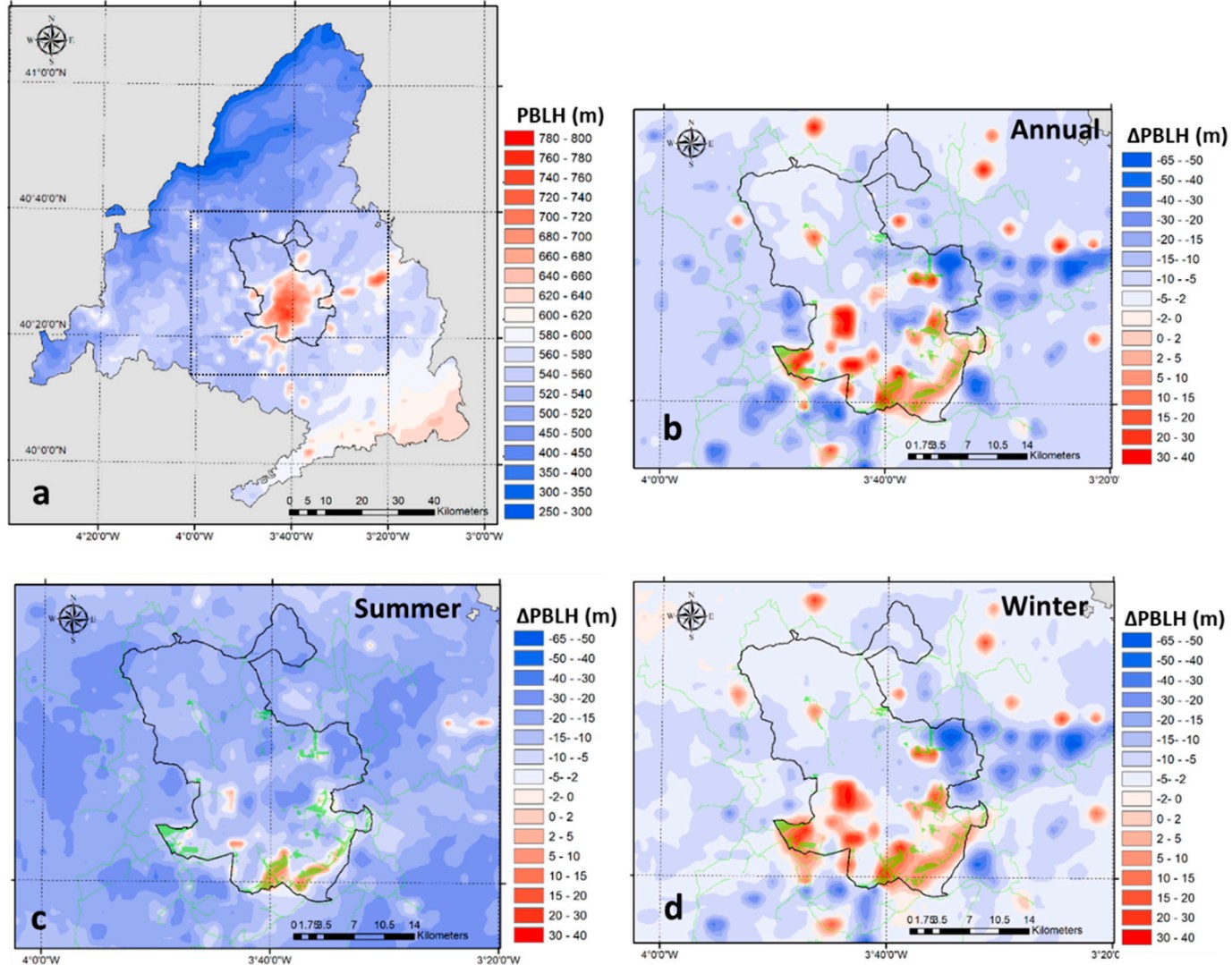

**Figure 6.** Annual planetary boundary layer height (PBLH) for the Baseline Scenario (**a**). Expected variation of PBLH: annual mean (**b**), summer mean (**c**) and winter mean (**d**), due to future vegetation.

Figure 6b shows contrasting results for the annual mean value and strong spatial gradients, from 250 m in the mountain range up to 800 m in downtown Madrid. This is consistent with previous studies that have reported PBLH variation in short spatial scales due to differential heating of different surfaces and land uses [71].

### 3.2. Impact of NBS on Air Quality

The assessment of the influence of future vegetation on air quality is based on the limit and target values defined by the Air Quality European Directive (2008/50/EC) for the protection of human health (Table S3).

Figure 7 shows the results for $NO_2$. This is a very important pollutant in Madrid since it has been the main target of previous pollution-control measures in both long- [55] and short-term [66] actions plans. Figure 7a shows the annual mean concentration for the Baseline Scenario (current situation). The highest concentration levels are found in downtown Madrid and in the main radial highways, where road traffic emissions [72] cause exceedances of the annual limit value for the protection of human health (40 $\mu g/m^3$). Figure 7b shows the expected changes in annual mean $NO_2$ due to the implementation of NBS in Madrid. Similarly to the results discussed for meteorological variables, the main differences between the Future and Baseline scenarios occur around the intervention areas (blue values imply concentration reductions, i.e., air-quality improvements, while red tones

indicate the opposite). We found local reductions in $NO_2$ average concentrations of up to 1.5 µg m$^{-3}$ (Table 3) (a 3–4% increase in relative terms, as illustrated in Figure S6) in those grid cells where temperature is increased (currently-treeless areas around the city). The increase in heat fluxes and, most importantly, turbulence intensity rise the mixing height, which contributes to reducing the concentration of this pollutant. Conversely, we found slight concentration increases (0.5%–0.75 µg m$^{-3}$, 2%–3%) elsewhere in the city, especially downtown Madrid ('Madrid Nuevo Norte'), where the future vegetation has a net cooling effect. It should be noted that additional sensitivity tests (not shown) confirm that concentration changes are mainly driven by the NBS impact on meteorological conditions and not so much for the increase of biogenic emissions. As expected, $O_3$ levels vary in the opposite direction (Figure 8b), since decreases in $NO_X$ imply less titration of NO and, thus, less consumption of $O_3$ to produce $NO_2$ [73].

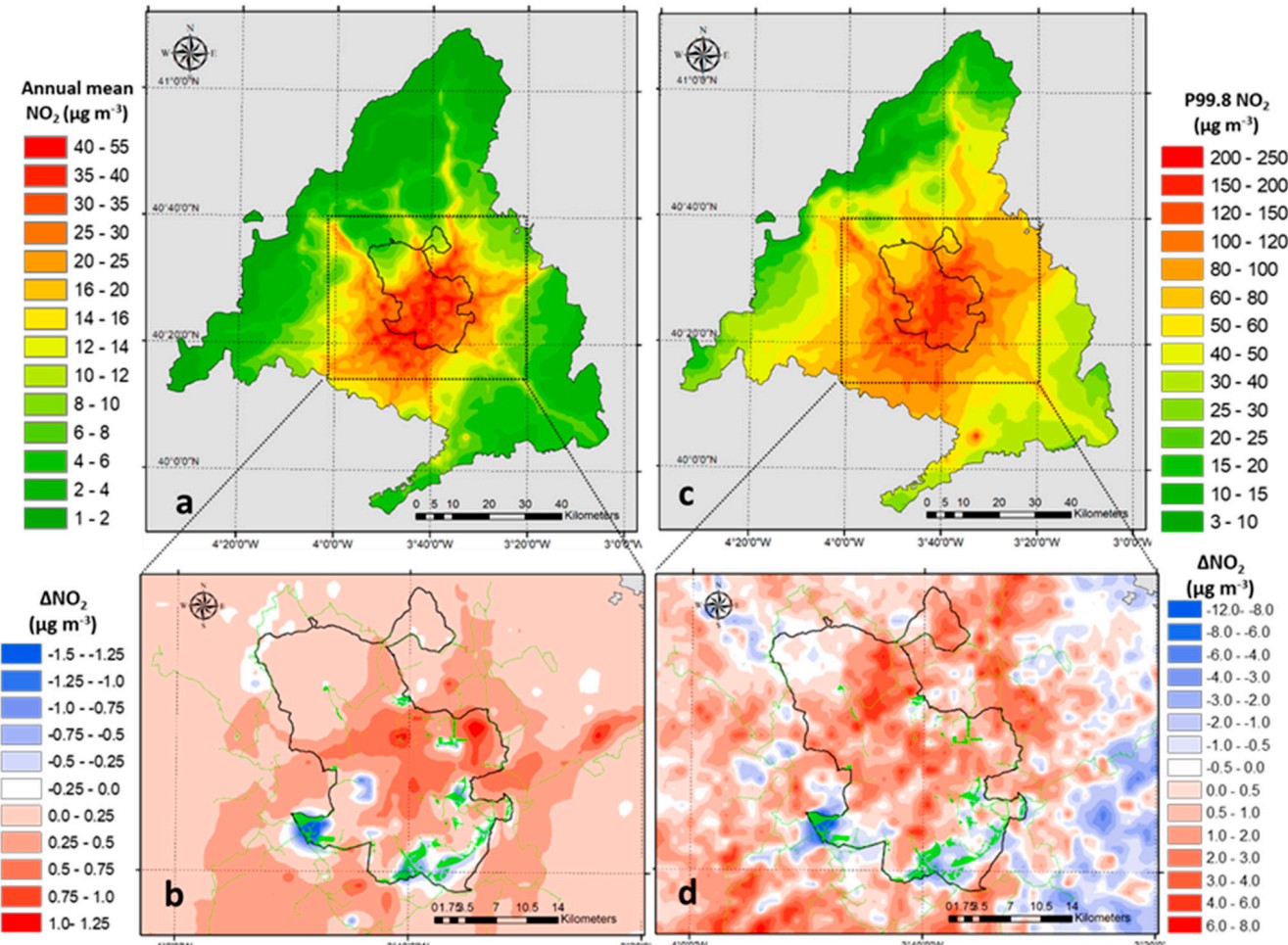

**Figure 7.** Annual mean of $NO_2$ (µg m$^{-3}$) for the Baseline Scenario (**a**) and expected variation (µg m$^{-3}$) due to future vegetation (**b**). Hourly 99.8 $NO_2$ percentile (µg m$^{-3}$) for the Baseline Scenario (**c**) and expected variation (µg m$^{-3}$) due to future vegetation (**d**).

**Table 3.** Summary of the effect of NBS on air pollution (relative change in brackets).

| Statistic | $NO_2$ (µg m$^{-3}$) | | $O_3$ (µg m$^{-3}$) | | $PM_{10}$ (µg m$^{-3}$) | | $PM_{2.5}$ (µg m$^{-3}$) |
|---|---|---|---|---|---|---|---|
| | **Annual Mean** | **P99.8** | **Annual Mean** | **P93.2** | **Annual Mean** | **P90.4** | **Annual Mean** |
| **Maximun** | 1.6, (4.7%) | 10.6, (14.1%) | 1.5, (3.2%) | 3.6, (2.9%) | 0.28, (2.0%) | 1.0, (6.0%) | 0.24, (3.4%) |
| **Minimun** | −1.5, (−4.1%) | −10.8, (−14.0%) | −1.7, (−4.3%) | −2.4, (−2.5%) | −0.30, (−1.7%) | −1.0, (−4.4%) | −0.25, (−2.4%) |
| **Average** | 0.1, (0.8%) | 0.4, (0.6%) | −0.04, (−0.1%) | 0.6, (0.5%) | 0.04, (0.3%) | 0.05, (0.3%) | 0.04, (0.7%) |

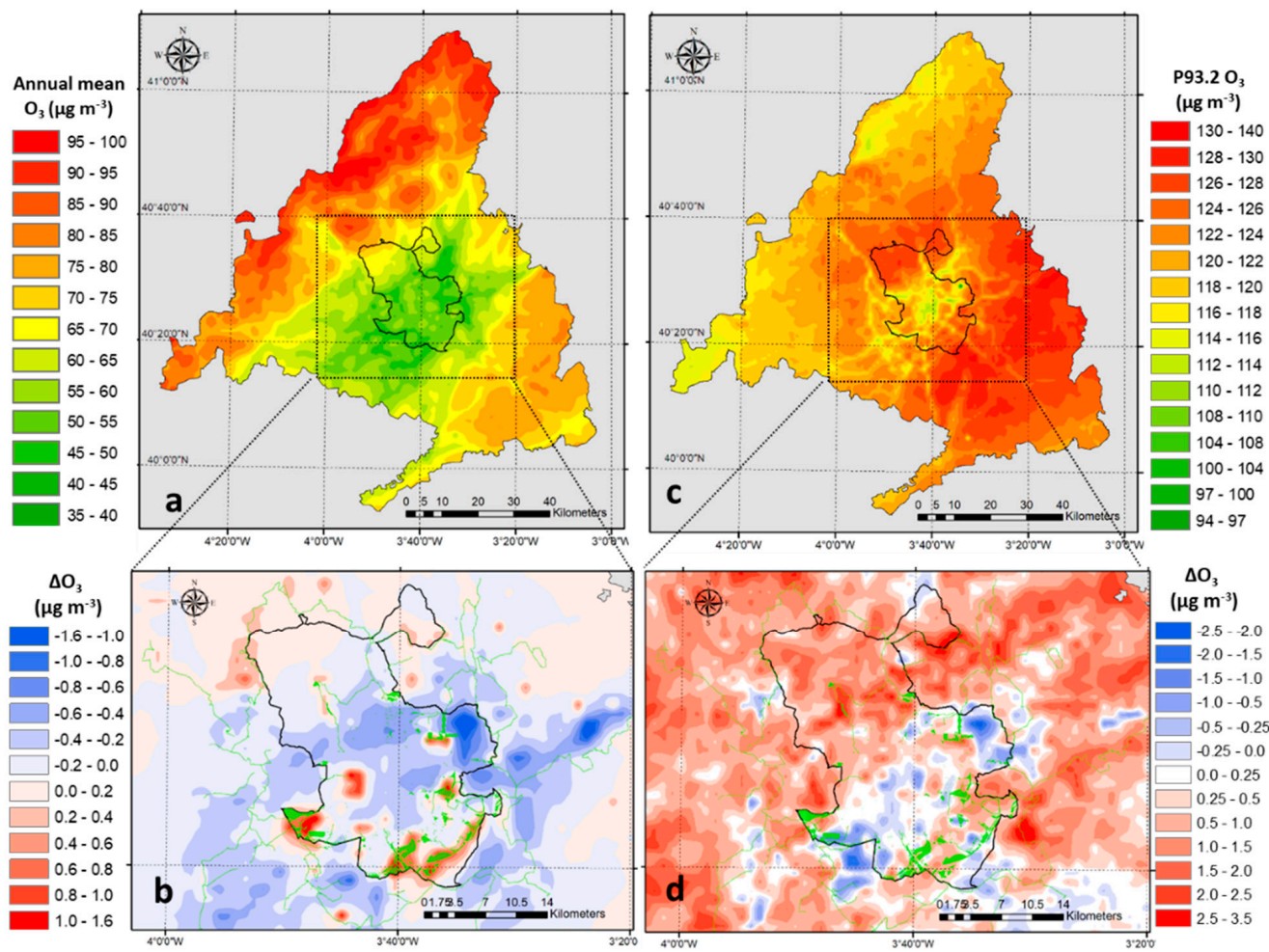

**Figure 8.** Annual mean of $O_3$ ($\mu$g m$^{-3}$) for the Baseline Scenario (**a**) and expected variation ($\mu$g m$^{-3}$) due to future vegetation (**b**). Daily percentile 93.2 of $O_3$ maximum daily eight-hour mean ($\mu$g m$^{-3}$) for the Baseline Scenario (**c**) and expected variation ($\mu$g m$^{-3}$) due to future vegetation (**d**).

As for $NO_2$ peak values, represented by the hourly percentile 99.8 (19th highest value within the calendar year), the spatial patterns are similar to those of the annual mean in the Baseline Scenario. In this case, we observe maximum values for the $NO_2$ hourly limit value close to 250 $\mu$g m$^{-3}$ (Figure 7c). Future vegetation (Figure 7d) would reduce these peaks in the range of 4–11 $\mu$g m$^{-3}$ (2%–8% according to Figure S6) in the more densely reforested areas of 'Bosque Metropolitano'. Increments of up to 5 $\mu$g m$^{-3}$ occur in some urban locations, presumably due to a decreased PBLH associated to the vegetation cooling effect within the city. A thinner mixing height may promote the accumulation of pollutants and give rise to slightly higher concentration levels [71,74].

Regarding changes in $O_3$ maximum daily eight-hour means (Figure 8d), slight increments of around 3 $\mu$g m$^{-3}$ (below 1%) are predicted in those locations where $NO_2$ peak values are expected to decline. Again, $O_3$ may be reduced in the city center, due to $NO_2$ increments. Contrarily to the effects found for the $O_3$ annual mean (not regulated), $O_3$ may marginally increase as well in non-urban areas, far from the envisaged NBS. This may be related to slower reactions involving VOCs [75]; although specific analyses would be needed to provide an accurate explanation.

The results for $PM_{10}$, summarized in Figure 9, show that current concentration patterns (Figure 9a,c) are strongly influenced by the prevailing NE-SW wind direction that disperse emissions from the city in that direction. As for the changes induced by future vegetation on this pollutant, maximum reductions of 0.3 $\mu$g m$^{-3}$ and 1.0 $\mu$g m$^{-3}$ (Table 3) (1%–2% and 2%–3%, respectively, as illustrated in Figure S7) are predicted for the annual

mean (Figure 9b) and the 36th highest 24-h mean (Figure 9d), respectively. The model anticipates some concentration increases within the city, specifically in the area affected by 'Madrid Nuevo Norte'. In this case, the substitution of urban fabric by natural cover leads to a cooling effect that contracts the PBL, causing slight increments of $PM_{10}$ ambient concentration (of around 0.2 µg m$^{-3}$, as an annual mean). It should be noted that both the $NO_2$ and PM expected increments are very small in relative terms (around 3% and 1%, respectively). The relatively smaller increment of PM ambient concentration may be related to a higher deposition rate that affects particulate matter more intensely.

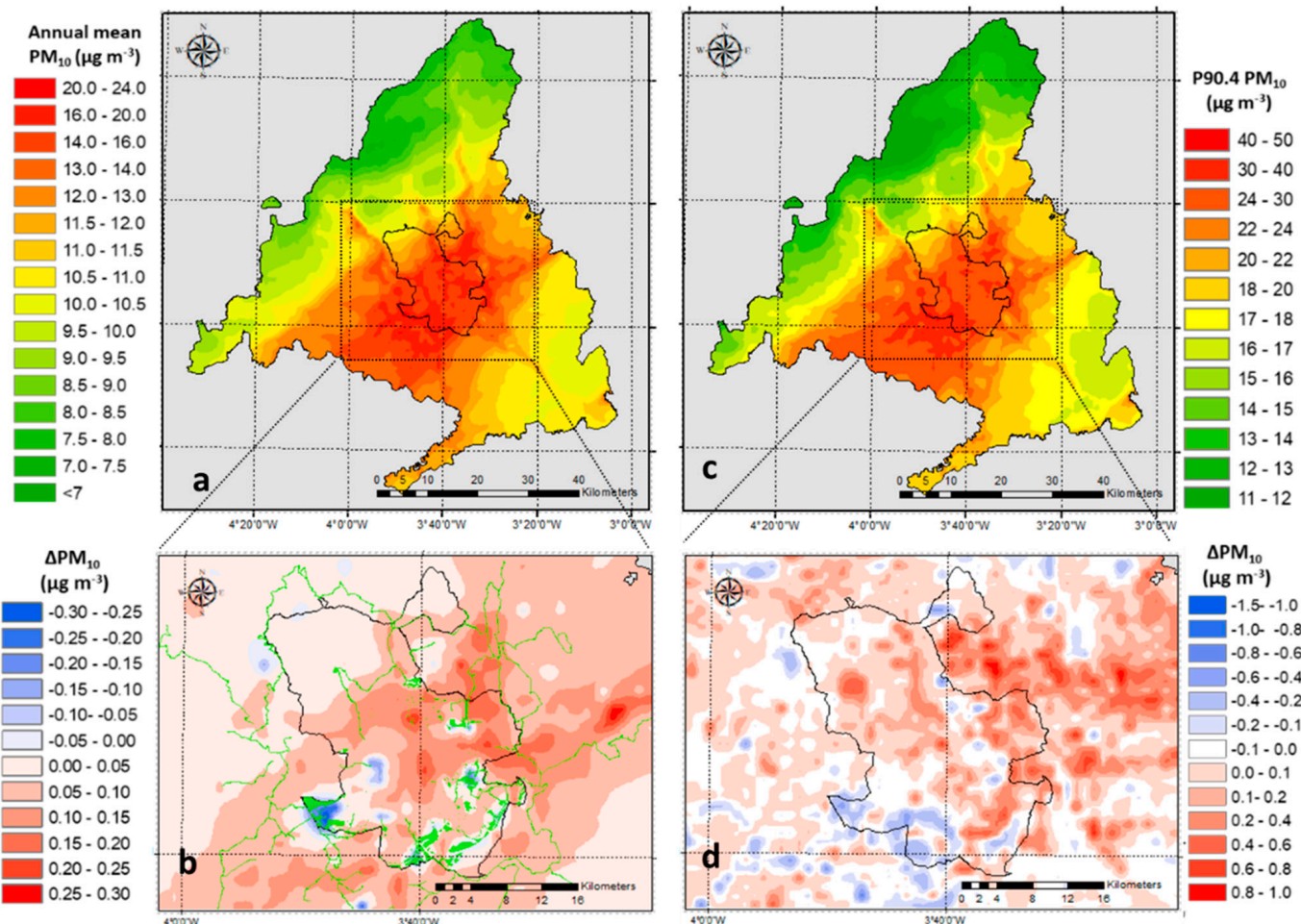

**Figure 9.** Annual mean of $PM_{10}$ (µg m$^{-3}$) for the Baseline Scenario (**a**) and expected variation (µg m$^{-3}$) due to future vegetation (**b**). Daily 90.4 $PM_{10}$ percentile (µg m$^{-3}$) for the Baseline Scenario (**c**) and expected variation (µg m$^{-3}$) due to future vegetation (**d**).

We found very similar results for finer particles ($PM_{2.5}$). As illustrated in Figure 10, very limited concentration decreases (of slightly higher than 0.2 µg m$^{-3}$ as an annual mean, approximately 2%) would occur in new wooden areas in the city outskirts. The combined effect of wind reduction and direction modification with reduced mixing height may result in generalized concentration increases in the metropolitan area of the city, if constant emissions are assumed.

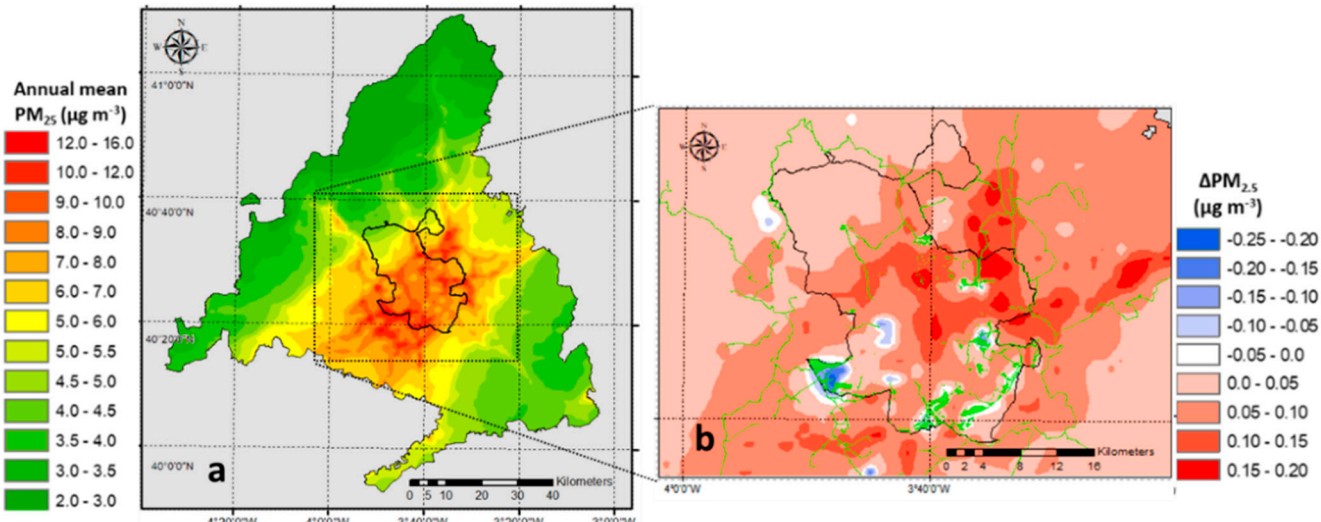

**Figure 10.** Annual mean of $PM_{2.5}$ ($\mu g\ m^{-3}$) for the Baseline Scenario (**a**) and expected variation ($\mu g\ m^{-3}$) due to future vegetation (**b**).

## 4. Conclusions

In this study, we describe and apply a methodology to assess the impact of urban vegetation at a city scale through a modelling approach. In particular, we investigate how future vegetation plans in Madrid may affect meteorology and air quality. Our results reveal that NBS may have a significant impact on important meteorological features, such as temperature, wind fields and planetary boundary layer height. Depending on the location of new vegetation, we found contrasting results. Interventions in consolidated urban areas ('Barrios productores' and 'Madrid Nuevo Norte') are expected to have a predominant cooling effect, while the introduction of vegetation in barren or scarcely vegetated areas in the city outskirts ('Arco Verde' and, mainly, 'Bosque Metropolitano') would induce average temperature increases. NBS generally increase friction, acting as a momentum sink and, thus, enhancing vertical mixing but reducing wind speed. However, we found that the impact on mixing height is dominated by thermal effects. As a result, under a constant-emission scenario, ambient concentration levels of $NO_2$ and PM would show moderate changes in the close vicinity of the interventions. Moderate changes are also predicted for $O_3$ concentrations due to future interventions, opposite in kind to those of $NO_2$. We also observe that NBS may have slight effects in relatively distant locations. In addition to the potential of vegetation to modify wind flows, this phenomenon strongly depends on local features (land uses and topography), so the impact of other interventions at a regional level cannot be anticipated.

Our findings suggest that vegetation may have significant effects locally, mainly due to its impact on urban meteorology, but it is essential to perform site-specific simulations from regional to local scales to take into account not only vegetation features but also other factors affecting regional atmospheric dynamics and chemical processes, such as land uses and topography. In addition, we conclude that vegetation may decrease pollutant concentration levels in some areas for some pollutants but also increase their concentration elsewhere. Therefore, it is necessary to combine NBS with further anthropogenic emission abatement measures. The proposed methodology could be used to assess the combined effect of NBS and emission-reduction measures within air-quality plans for urbanized areas.

**Supplementary Materials:** The following supporting information can be downloaded at: https://www.mdpi.com/article/10.3390/f13050690/s1, Table S1. Dimensions and spatial resolution of WRF and CMAQ modelling domains; Table S2. Air quality parameter regulated by the Directive 2008/50/EC on ambient air quality and cleaner air for Europe; Table S3. Properties of relevant natural-land uses (WRF VEGPARM file); Table S4. Statistics: NOAA meteorological stations; Table S5.

Statistics: Madrid meteorological stations; Figure S1. Annual mean temperature, wind speed (10 m) and PBL height relative changes (100 × (Future − Baseline) / Baseline); Figure S2. Seasonal mean variation (Future − Baseline) sensible heat flux (W m$^{-2}$) at the surface in summer (a) and winter (b). Seasonal mean variation (Future − Baseline) latent heat flux at the surface in summer (c) and winter (d); Figure S3. Seasonal mean variation (Future − Baseline) of solar radiation absorbed at ground in winter (a) and summer (b); Figure S4. Detailed topographic map of the study area; Figure S5. Seasonal mean variation (Future − Baseline) of friction velocity (ΔUSTAR) and wind speed in summer (a) and winter (b). Seasonal mean variation (Future − Baseline) wind direction in summer (c) and winter (d); Figure S6. $NO_2$ and $O_3$ relative changes (100 × (Future − Baseline) / Baseline); Figure S7. $PM_{10}$ and $PM_{2.5}$ relative changes (100 × (Future − Baseline) / Baseline).

**Author Contributions:** Conceptualization: D.d.l.P., J.M.d.A., A.N., C.S., S.F. (Sandro Finardi), S.F. (Silvano Fares), L.T., R.B. and M.M.; Methodology: D.d.l.P., J.M.d.A., R.B. and M.M.; Software: D.d.l.P. and C.S.; Validation: A.N., S.F. (Sandro Finardi) and S.F. (Silvano Fares); Formal analysis: D.d.l.P., J.M.d.A. and R.B.; Investigation: D.d.l.P., J.M.d.A., A.N., C.S., S.F. (Sandro Finardi), S.F. (Silvano Fares), L.T., R.B. and M.M.; Resources: D.d.l.P., C.S. and S.F. (Sandro Finardi); Data Curation: D.d.l.P., J.M.d.A., L.T., R.B. and M.M.; Writing—original draft preparation: D.d.l.P., J.M.d.A. and R.B.; Writing—review and editing: D.d.l.P., J.M.d.A., A.N., C.S., S.F. (Sandro Finardi), S.F. (Silvano Fares), L.T., R.B. and M.M.; Visualization: D.d.l.P.; Supervision: R.B.; Project Administration: M.M.; Funding Acquisition: S.F. (Sandro Finardi), S.F. (Silvano Fares), L.T., R.B. and M.M. All authors have read and agreed to the published version of the manuscript.

**Funding:** This study was developed under the project Life VEG-GAP (https://www.lifeveggap.eu/) Accessed 20 April 2022) which was funded by European Union Life Program in 2018, Grant Number LIFE18 PRE IT 003.

**Data Availability Statement:** Data available in Information platform: https://veggaplatform.enea.it (accessed on 27 April 2022).

**Acknowledgments:** We acknowledge Madrid's Air Quality Service for providing meteorological dataset from the City Hall meteorological network.

**Conflicts of Interest:** The authors declare no conflict of interest.

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
