# Peer review of "Assessment of Air Quality and Meteorological Changes Induced by Future Vegetation in Madrid"

_forests, doi:10.3390/f13050690_

Round 1

Reviewer 1 Report

This work presents a methodology to assess the impact of future vegetation scenarios on meteorology and air quality at city scale, using the WRF-BEP-CMAQ modelling system. The methodology is applied in the Madrid area considering the current NBS strategies foreseen for the next decades. The manuscript is very well written and the methodology is adequately described. I only recommend some minor changes for publication.

Methods: I would suggest to simplify the description of the land use classes and their modifications for the base and future scenarios. Please, revise legends and colors in Figure 2 and description of Figure 3. In the last paragraph of page 6, land use classes are changed.

Figures 4 to 9: It would be useful for the reader to see some measure of the relative impacts of the differences when discussing each figure. These are mentioned qualitatively in the conclusions (e.g., in L402 and L412) but not in the results. On the other hand, the largest differences obtained in winter and in the annual mean values in Figures 4 to 6 seem to have similar magnitudes. Could you comment on why this is?

L248 and 254: Replace “Figure 3” with “Figure 4”.

L259: “both positive and negative”. In Figure 3, this is mainly observed with the positive differences.

Author Response

Please, find our point-to-point responses in "Response to Reviewer 1_Forests_1680917.pdf"

Reviewer 2 Report

This manuscript focused on the numerical scenario simulations of future changes of vegetation via the WRF-BEP-CMAQ model based on the former work of Paz et al. (2016). In the research area of city heat-island effects, several similar papers had been published, but the related articles are lacked in the introduction and discussion parts.

Major questions:

  1. The new added urban land uses are not well introduced in the part of methodology.
  2. At the end of the results and discussion, an overall summary table needed to be added with presenting the quantitative effects of each meteorological and air quality variables, and this will make the readers easy to get the take-home messages.
  3. Several inappropriate self-citations are found in this manuscript, such as L309, L330 and so on. In these sentences, the citations are not needed as they are the major results of this study.
  4. Why the RH is not analyzed as it has significant effects on air quality? And the conclusion part need to be shorten.

The whole manuscript is presented in high quality but still has many minor questions as follows:

  • Line 131: The comma symbol is lacked at the end of 2015.
  • Line 134-135: Pay attention to English grammar.
  • Line 193: in or simulation?
  • Line 229: % should be percentage.
  • Line 243: a new table need to be added as former stated.
  • Line 29, 272, 289, 294, 297 and so on: the word of ‘observed’ should be carefully used in model simulation manuscript.
  • Line 276: lacking the symbol of full stop.
  • Line 280: ‘Averaged changes’ should be ‘Yearly averaged changes’.
  • Line 300:also indicates.
  • Line 301: an underline is presented under the symbol of degree.
  • Line 319-320: what is the annual value?
  • Line 331: found -> found at.
  • Line 344-346: The result of sensitivity tests should be provided in the supplementary file to support this conclusion.
  • Line 363: ‘were’ should be ‘where’.
  • Line 386: ‘moderate’ should be ‘very limited’, as the word of ‘moderate’ had been used for NO2.
  • Line 410: ‘observed’ is easy to misleading the readers.
  • Line 413: it is a common sense.
  • Line 422: ‘improve’ should be ‘decrease’.

Author Response

Please, find our point-to-point responses in "Response to Reviewer 2_Forests_1680917.pdf"

Reviewer 3 Report

In the manuscript “Assessment of air quality and meteorological changes induced by future vegetation in Madrid” by de la Paz et al., the authors conducted numerical simulations to investigate the impacts of vegetations changes on 1) urban meteorology, and 2) air quality variables (NO2, O3, and PM) in Madrid. The authors found non-trivial impacts induced by changing the vegetation, noting e.g. a net heating effect when introducing trees to the outskirts of the city, whereas the authors found a net cooling effect when the trees were more consolidated within urban areas. The aim of the study is clear, and the research design and methods are appropriate. Before the manuscript can be published, the following items need to be addressed.

Major comments:

  • Although the authors include a table in the supplementary materials summarizing the different parameterization schemes used in their modeling framework, it would be helpful to the reader to include this material within the body of the manuscript body itself. The authors also need to justify their choice for these certain parameterization schemes, especially those most relevant to their study i.e. the Boulac PBL scheme.
  • To have confidence in the model output, comparisons between the simulations and observations of the surface layer and boundary layer within in the region are critical to gain fidelity in the model results presented.

Minor comments

  • Font size is inconsistent in the “Supplementary Materials” section starting on Line 428.
  • The references need to be formatted consistently. For example, sometimes “https” appears before the DOI, whereas other times it does not. The link to the DOI also appears in blue for some, but not all, citations.

Line-specific comments

  • Line 128: What is the lowest model level?
  • Line 131: More details are needed as to why the year 2015 was chosen.
  • Line 142-144: Are the vegetation differences the only thing that is changed between the two sets of simulations, and are both simulations run for an entire year?
  • Line 134-135: Do you mean “and O3”?
  • Line 218: The y-axis in the lower part of Figure 3 is obscured by the graph title. Furthermore, the y-axis needs to be labeled and with units.
  • Line 276: Missing period
  • Line 298: “wind speed” is two words.
  • Line 407: Rephrase “would turn out in”
  • Line 415: No comma after “phenomenon.”

Author Response

Please, find our point-to-point responses in "Response to Reviewer 3_Forests_1680917.pdf"

Round 2

Reviewer 3 Report

The authors have satisfactorily addressed my original concerns about the manuscript. I recommend that it can be published in its present form.